# Comprehensive Cortical Structural Features Predict the Efficacy of Cognitive Behavioral Therapy in Obsessive-Compulsive Disorder

**DOI:** 10.3390/brainsci12070921

**Published:** 2022-07-14

**Authors:** Xiang-Yun Yang, Rui Liu, Jia Luo, Fang-Fang Huang, Peng-Chong Wang, Xiao-Jie Yang, Hang Wu, Yuan Zhou, Zhan-Jiang Li

**Affiliations:** 1Department of Clinical Psychology, Beijing Anding Hospital, Capital Medical University, Beijing 100088, China; wysun828@126.com (X.-Y.Y.); liuruicomeon@163.com (R.L.); luojia529@aliyun.com (J.L.); hff87hff87@163.com (F.-F.H.); pengchongpsy@163.com (P.-C.W.); xiaojie222yabin@163.com (X.-J.Y.); 13500347874@163.com (H.W.); 2The National Clinical Research Center for Mental Disorders & Beijing Key Laboratory of Mental Disorders, Beijing Anding Hospital, Beijing 100088, China; 3CAS Key Laboratory of Behavioral Science & Magnetic Resonance Imaging Research Center, Institute of Psychology, Chinese Academy of Sciences, Beijing 100049, China; 4Department of Psychology, University of Chinese Academy of Sciences, Beijing 100049, China

**Keywords:** obsessive-compulsive disorder, cognitive behavioral therapy, cortical structural feature, prediction, machine learning, support vector regression

## Abstract

Although cognitive behavioral therapy (CBT) is effective for patients with obsessive-compulsive disorder (OCD), 40% of OCD patients show a poor response to CBT. This study aimed to identify the cortical structural factors that predict CBT outcomes in OCD patients. A total of 56 patients with OCD received baseline structural MRI (sMRI) scanning and 14 individual CBT sessions. The linear support vector regression (SVR) models were used to identify the predictive performance of sMRI indices, including gray matter volume, cortical thickness, sulcal depth, and gyrification value. The patients’ OC symptoms decreased significantly after CBT intervention (*p* < 0.001). We found the model with the comprehensive variables exhibited better performance than the models with single structural indices (MAE = 0.14, MSE = 0.03, R^2^ = 0.36), showing a significant correlation between the true value and the predicted value (r = 0.63, *p* < 0.001). The results indicated that a model integrating four cortical structural features can accurately predict the effectiveness of CBT for OCD. Future models incorporating other brain indicators, including brain functional indicators, EEG indicators, neurotransmitters, etc., which might be more accurate for predicting the effectiveness of CBT for OCD, are needed.

## 1. Introduction

About 1–3% of the population suffer from obsessive–compulsive disorder (OCD) once in their lifetime [1,2]. OCD patients suffer from persistent intrusive thoughts (obsessions) and/or repetitive behaviors (compulsions), which cause social disorders that are expensive to treat [3]. Through altering patients’ dysfunctional thinking and behavior, cognitive behavioral therapy (CBT) may improve the OC symptoms and negative emotions. CBT has become one of the first-line treatments for OCD, with its response rate reaching 60–70% [4,5]. Compared with pharmacotherapy, CBT can reduce relapse rates and is associated with fewer adverse side effects. Despite these benefits, the unclear OCD pathology and lack of treatment response predictors limit the application of CBT for OCD treatment. At least 40% OCD patients respond poorly to CBT. It is difficult to judge which patients will respond better to CBT, based only on the experience of clinicians. Magnetic resonance imaging (MRI), EEG, and positron emission tomography are non-invasive technologies that conveniently acquire detailed brain data. Moreover, by combining these with advanced computing methods, the brain imaging data can be utilized to uncover the pathology underlying mental disorders, as well as reveal the treatment targets and brain biological markers that can predict the treatment outcome [6].

Using the structural MRI (sMRI) technique, numerous studies have confirmed structural abnormalities in many brain regions of OCD patients, including reduced gray matter volumes in the prefrontal, cingulate, thalamic, and temporal limbic regions [2]. Insight into neurobiological mechanism can uncover biological indicators for predicting the treatment outcome of CBT for OCD. Some studies have shown that CBT might volumetrically affect the key brain regions, including the orbitofrontal cortex (OFC) [7,8], striatum [7,9], and thalamus [7], that are involved in the neuroanatomy of OCD. Furthermore, a study comparing the gray matter volume (GMV) between two groups of OCD patients responding differently to CBT showed that non-remission OCD patients have a significantly smaller GMV in the left dorsolateral prefrontal cortex (DLPFC) than the remission group [10], implying that specific brain structural changes are associated with better CBT outcome in OCD patients.

Although the above studies have demonstrated that GMV changes were associated with the efficacy of CBT, no study has explored whether the baseline GMV predicts CBT’s efficacy for OCD using a rigorous prediction model. The machine learning method comprises a large set of statistical learning algorithms, and its application in neuroimaging data provides a powerful tool for brain studies. This advanced statistical method has been widely used to identify predictors of various treatment methods [11]. Identifying biological characteristics of patients with OCD that can predict treatment outcomes can enhance patient selection to reduce wasteful treatment.

Previous OCD studies utilizing sMRI mainly focused on GMV alteration. Recently, the surfaced-based analysis has been shown to increase the accuracy of brain registration [12]. Studies have shown that the differentiation of the brain surface into gyrification and sulci is related to a variety of neurodevelopmental disorders. For example, local gyrification has been linked to certain important cognitive functions, including working memory, mental flexibility tasks, as well as attention and semantic verbal fluency tasks in a healthy population [13,14]. Researchers explored the gyrification in OCD patients and found inconsistent alterations in cortical folding in OCD. The majority of studies reported hypogyrification [15,16], or gender-specific hypogyrification [17] exists in OCD patients, and one study reported hypergyrification [18]. Although the number of studies on different cortical structural indices is limited, the existing findings indicate that including the multi-dimensional cortex enhances the understanding of the pathological mechanism and therapeutic targets of OCD.

Based on the cognitive functions of GMV, cortical thickness, sulcal depth and gyrification, this study aimed to explore the effect of these cortical structural characteristics on predicting CBT efficacy for OCD. The predictive performance of sMRI indices, including GMV, cortical thickness, sulcal depth, and gyrification, and the comprehensive variables containing these four indicators was analyzed using the linear support vector regression (SVR) model. We hypothesized that the integrated structural index is more than a single index in predicting the treatment outcome of CBT in OCD patients. To test this hypothesis, we recruited 60 patients with OCD who would receive baseline sMRI scanning. Each individual underwent 14 CBT sessions. Five treatment outcome prediction models were constructed using an SVR model. Change in OC symptoms was the dependent variable, whereas and the selected cortical structural features were the independent variables.

## 2. Materials and Methods

### 2.1. Participants

A total of 60 outpatient OCD individuals were recruited. They receive CBT intervention, clinical evaluation, and MRI scanning at the Beijing Anding Hospital, Capital Medical University, between January 2013 and October 2016. OCD diagnosis was based on the Diagnostic and Statistical Manual of Mental Disorders, Fourth Edition (DSM-IV) guidelines. To be eligible, one had to have fulfilled the following: (1) being right-handed; (2) aged between 18 and 50 years; (3) with a Yale-Brown Obsessive–Compulsive Scale (Y−BOCS) score of 16 or more; (4) with a score of less than 18 in the 17-item Hamilton Depression Rating Scale (HAMD−17); and (5) having never received a psychiatric medication or psychological therapy or stopped medication for at least 4 weeks before enrollment. To avoid the possible effects of comorbid disorders, participants with neurological illness, major physical illness, or other psychiatric disorders were excluded. In addition, this study also excluded participants who were pregnant or had any MRI contraindications.

This study was approved by the Research Ethics Committee of Beijing Anding Hospital, Capital Medical University (approval code (2013) Keyan (7)). All study participants consented to participate in this study in writing.

### 2.2. CBT Treatment

All of the OCD patients underwent 14 sessions of the CBT program, which mainly included exposure and response prevention (ERP) and cognitive therapy. The CBT was performed as previously described (Yang et al., 2015) [5] by four well-trained CBT therapists. During the treatment process, every CBT therapist accepted one-hour supervision per week from a senior therapist to ensure the treatment quality. The CBT consisted of 14 sessions over 12 weeks, with each session lasting 60 min. The 14 sessions of the whole treatment are briefly described in Table 1.

### 2.3. Diagnosis and Symptom Evaluation

Four experienced senior psychiatrists not involved in the CBT program performed the diagnosis and symptom evaluation. Prior to the study, the four evaluators participated in joint training sessions for diagnosis and scale assessment. The evaluators screened all participants and performed OCD diagnosis using the Structured Clinical Interview for DSM-IV Axis I Disorders tool. The severity of obsessive–compulsive symptoms, depression, and anxiety were assessed using the Y−BOCS, 17-item Hamilton Depression Rating Scale (HAMD−17), and the Hamilton Anxiety Scale (HAMA). The assessments were performed separately before and after 14 treatment sessions.

The main efficacy indicator of this study was the improvement of obsessive–compulsive symptoms. Thus, the treatment effect was based on the percentage decrease ((score *_pretreatment_* − score *_posttreatment_*)/score *_pretreatment_* × 100%) in the Y−BOCS total score. 

The change of HAMA and HAMD before and after CBT were secondary CBT efficacy indicators.

### 2.4. MRI Data Acquisition

All the participants were assessed using a SIEMENS 3.0 T Trio scanner (TIM Systems) using a 32-channel head coil, at the State Key Laboratory of Cognitive Neuroscience and Learning, Beijing Normal University, China, before the start of treatment. Participants laid supine and were instructed to be still, relax, keep their eyes closed, avoid falling asleep and not think of anything in particular during scanning. Head motion and scanner noise were minimized using foam pads and earplugs, respectively. T1-weighted images were acquired using a sagittal 3D magnetization prepared rapid gradient echo (MP-RAGE) sequence with the following features: TR/TE = 2530/3.39 ms; FOV = 256 × 256 mm^2^; in-plane resolution = 256 × 256; time to inversion (TI) = 1100 ms; flip angle = 7°; slice thickness = 1.33 mm, voxel size = 1 mm × 1 mm × 1.33 mm. In addition, there were 144 interleaved sagittal slices covering the whole brain. 

### 2.5. MRI Data Preprocessing

Preoperative MRI data was preprocessed using the standardized pipeline of the CAT12.8 toolbox (r1872, http://dbm.neuro.uni-jena.de/cat12/ accessed on 20 August 2021) for SPM12 (v7487, https://www.fil.ion.ucl.ac.uk/spm/software/spm12 accessed on 14 November 2018) in the MATLAB programming language (R2018b). First, structural image data were segmented into gray and white matter as well as cerebrospinal fluid, according to a prior tissue probability map (TPM) provided by the CAT12 toolbox. These three compartments were adopted to calculate the total intracranial volume in native space. Afterwards, all the native-space tissue segmentations were registered to a standard Montreal Neurological Institute (MNI) template. To mitigate volume changes that can be caused by spatial normalization, images were modulated to preserve the total amount of gray matter. Finally, the segmented gray matter and white matter maps were smoothed using a Gaussian kernel with 8 mm full width at half maximum (FWHM).

### 2.6. Statistical Analysis

#### 2.6.1. Clinical Data Analysis

Grouped data, including age, gender, education level, and illness duration, were described using descriptive analysis. The pre-to-post comparisons of clinical symptoms (Y−BOCS, HAMD and HAMA scores) in OCD patients were performed using paired−sample *t*-tests. Data were analyzed using the SPSS software, version 24.0. The statistical significance level was set as 0.05 (two-tailed).

#### 2.6.2. Extraction of Structural Features and Analysis of the Prediction Models 

After MRI data preprocessing, a total of 377 structural features were extracted based on the atlas. Among of them, 170 GMV were extracted from the automated anatomical labelling 3 (AAL3) atlas [19]; 69 cortical thickness, sulcal depth, and gyrification values were obtained from the Desikan–Killiany atlas (DK40), [20] (Figure 1). For each structural feature, a linear regression method was conducted to regress out the effect of age, gender, and education. Further, to extract the features, a Pearson correlation was computed between the Y−BOCS reduction rate and all the structural features including GMV, cortical thickness, sulcal depth, and gyrification values. The features were ranked from maximum to minimum according the correlation values; then, the top ten features were selected to construct the prediction model. Thus, 10 comprehensive variables were selected from all the structural features, and 10 variables were selected from the GMV, cortical thickness, sulcal depth, and gyrification values, respectively.

Five prediction models were constructed using a linear SVR model with the leave-one-out cross validation (LOOCV) method based on the selected features, using GMV, cortical thickness, sulcal depth, gyrification value, and comprehensive variables. The reduction rate of Y-BOCS score at 12 weeks (reduction rate = (Y−BOCS score at baseline − Y−BOCS score at 12-week)/Y−BOCS score at baseline) was the dependent variable, whereas the selected features were the independent variables. To estimate the model performance, the mean absolute error (MAE), mean squared error (MSE), coefficient of determination (R^2^) and Pearson correlation (R) were calculated. Furthermore, permutation tests were performed 1000 times to verify the generalizability of the model. The whole process of the model building was performed in Python using the scikit-learn toolkit [11].

Further, to explore the correlation between the reduction rate of Y−BOCS and the features with best performance of the prediction model, the Pearson correlation method was conducted.

## 3. Results

### 3.1. Demographic and Clinical Characteristics of OCD Patients

A total of 60 patients with OCD were recruited in this study. Of these, four patients withdrew from the study due to time inconvenience and transport challenges (less than six CBT sessions). In total, 56 patients underwent the 14 CBT sessions and received two-timepoint symptom evaluation. The demographic and clinical characteristics of the 56 patients were shown in Table 1. The majority of the participants were male (66.07%). Save for one patient, the study participants ranged from 20 to 45 years old. The patients’ obsessive–compulsive symptoms were moderate or higher, accompanied by mild anxiety and depressive symptoms. We compared the change of scores on Y−BOCS, HAMD, and HAMA after CBT. As shown in Table 2, there were significant differences in the Y−BOCS, HAMD, and HAMA scores before and after CBT among the OCD patients (all *p* < 0.001).

### 3.2. The Prediction Models Analysis Based on the Structural Features in OCD Patients

As shown in the Table 3, each of top ten features in the GMV, cortical thickness, sulcal depth, and gyrus value were extracted in this study, individually. In addition to clarifying the individual prediction power of these four single structural indices, we also paid attention to the comprehensive prediction effect of these four features, that is, comprehensive variables. The results of five prediction model analyses showed the model with the comprehensive variables exhibited a better performance than a single structural indices (MAE = 0.14, MSE = 0.03, R^2^ = 0.36). As shown in Table 3 and Figure 2, the comprehensive variables included GMV in the left cerebellar hemisphere 10 (lCER10) and the bilateral ventral tegmental area (lVTA and rVTA), the cortical thickness in the right entorhinal cortex (entorhinal), the gyrification value in the right isthmus cingulate, and the sulcal depth of left superior frontal, left fusiform, left lateral occipital, left parahippocampal, and left rostral middle frontal regions.

Furthermore, based on the results of prediction models, we analyzed the correlation between the true value and predictive value of each feature to verify the performance of the models. As shown in Figure 3, the comprehensive variables had a significant correlation between the true value and predictive value (r = 0.63, *p* < 0.001).

### 3.3. The Post Hoc Analysis of the Prediction Model with the Comprehensive Variables

In order to further clarify the association between each component in the comprehensive variables and OC symptom change, we applied a correlation analysis to analyze the correlation between 10 features and the reduction rate of the Y−BOCS score. As shown in Figure 4, seven components had a significant correlation with the reduction rate of the Y−BOCS score. Due to the gender differences in the participants, we also compared the comprehensive variables between male patients and female patients. The two sample *t*-test showed that there were significant differences in the gray matter volume of the left cerebellum 10 (*p* = 0.006) and the sulcal depth of the left parahippocampal (*p* = 0.029) between the two groups.

## 4. Discussion

This study sought to explore the predictive effect of cortical structural features for the efficacy of 12 weeks of CBT on OCD using the SVR model. To the best of our knowledge, this is the first study to compare the predictive performances between a single structural feature and a combination of several features. The results confirmed our hypothesis that a model integrating four structural features is more accurate than single feature models. The structural features included sulcal depth (five regions), GMV (three regions), cortical thickness (one region), and the gyrus value (one region). Current preliminary findings suggest that neuroimaging-based computer-aided methods can effectively predict OCD patients who would benefit from CBT. 

### 4.1. The Efficacy of CBT for OCD Patients

To explore the specific action of CBT for OCD patients, the current study excluded OCD patients with comorbidities. All patients received a single CBT program without taking any drugs or other types of treatment. The paired-*t* test found significant improvements in OC symptoms, and depressive and anxiety symptoms among OCD patients after CBT. In all, 46 patients (82.14%) had a ≥35% reduction in their Y−BOCS total score and were considered responsive to CBT. The results are consistent with other previous studies, indicating that CBT is an effective treatment for OCD [4,5].

Although the group-level analysis showed CBT significantly alleviated OCD after 12 weeks, 10 patients were unresponsive to the treatment (Y−BOCS total score reduction rate <35%). Thus, it is imperative to identify biological factors that can predict CBT outcomes in OCD patients. This can promote the development of individualized treatment plans for those patients, which improves the treatment outcome and saves on the wastage of resources.

### 4.2. The Performance of Five Prediction Models on CBT in OCD Patients

In this study, the GMV, cortical thickness, gyrification value, and sulcal depth were considered as sMRI features in our algorithm. We constructed five prediction models using SVR based on four selected features and the comprehensive variables, respectively. Our results demonstrated that cortical structural features may predict which patients are likely to respond to CBT. It is particularly noteworthy that comprehensive features showed a higher predictive weight in all features, which proved our hypothesis. The formation of cortical morphology such as the sulci and gyrus, are involved in the development of brain function. It is reported that thickness, sulci and gyration are affected by different genetics and have different tracks in the process of brain development [21]. This difference highlights the importance of exploring the different cortical structural features separately.

CBT can alter an individual’s dysfunctional thinking, emotion, and behavior through cognitive and behavioral techniques, so as to improve obsessive–compulsive symptoms. In addition to the psychological mechanism, studies have confirmed that CBT has a neurobiological mechanism in the treatment of OCD [5,22,23]. Therefore, it is feasible to look for brain imaging indicators to predict the efficacy of CBT in the treatment of OCD. Considering the complexity of brain structure and function, the inclusion of comprehensive indicators is helpful to improve the predictive power. Although no study has compared the predictive effect between single and integrated cortical structure for CBT, several neuroimaging studies have indicated that integration of brain features have a better identification power of the neuropsychiatric disease, such as the integration of sMRI and fMRI [24], and the integration of GM and white matter [25]. Consistent with these studies, our finding proved that comprehensive cortical structural features had a better predictive performance than a single feature for CBT in the treatment of OCD.

### 4.3. Key Cortical Structural Feature and Regions for Predicting the Efficacy of CBT in OCD

As noted, this study found the comprehensive cortical structural features had a better predictive performance than a single feature. In the comprehensive variable prediction model, the sulcus depth in five regions and the gyrification in one region were correlated with the improvement in OC symptoms. This result suggests that the gyrification has a higher predictive weight on CBT efficacy in OCD patients. Gyrification, the process leading to the characteristic differentiation of the brain’s surface into sulci and gyri, is associated with cortical maturation and closely linked to neurodevelopment [26]. Studies indicated that gyrification is related to the integrity of the cortex and subcortical circuitry of the developing brain.

We found the sulcus depth of five brain regions, including the left superior frontal, left rostral median frontal, left lateral occipital, left parahippocampal and left fusiform, regions, which are critical in the neuro-mechanism of OCD, were related to the CBT outcome. Recent neuroimaging studies suggest that both orbitofronto–striatum circuits and prefronto–limbic–posterior circuits constitute a complicated pathophysiological state of OCD [2,27]. It was worth noting that the findings show obvious left laterality. The complex functional organization of the human brain is characterized by left–right asymmetry [28]. Altered functional and structural laterality have also been investigated in OCD, especially in the visuospatial domain [29,30], as well as altered emotional processing [31]. Furthermore, a study exploring the effect of transcranial magnetic stimulation in treatment-resistant OCD revealed that the abnormal hemispheric laterality was normalized after the treatment [32]. In the future, studies should explore whether CBT induces different changes in the left and right hemispheres.

The GMV in the left cerebellar hemisphere and bilateral ventral tegmental area (VTA) was found to have a significant role in the predicting effect for CBT. Most sMRI studies on OCD proved the important role of GMV alterations in the neuro-mechanism of OCD. Recent studies indicated that the cerebellum has become an important pathological region except the classical cortico–striato–thalamo–cortical circuitry in OCD [27]. Our previous study found that CBT can alter the regional homogeneity of the left cerebellum in OCD, and the percentage change of regional homogeneity was positively correlated with the percentage reductions in Y-BOCS compulsion scores [5]. Current findings suggest that cerebellum may be an important biomarker for OCD, and has the potential effect to predict CBT efficacy.

The VTA, which contains dopaminergic neurons and GABAergic neurons, is a critical part in the cortico–striatal–ventral tegmental network related to repeated behavior [33]. It projects a chemically diverse signal to regions such as the prefrontal cortex, amygdala, and ventral striatum [34]. These anatomical characteristics enable the VTA to subserve a diverse assortment of cognitive functions in the downstream brain regions. The VTA is a major source of dopamine input to ventral striata, and processes reinforcement signals, which inform behavioral selection. The pathway from the VTA to the striatum is classically associated with producing and sustaining effortful behaviors, mediating locomotor arousal, and energizing behavioral responses [35]. The disruptions in this pathway would theoretically impact information processing necessary for selecting and sustaining behavioral sequences, and could promote compulsive behavioral patterns. Exposure and response prevention (ERP) is an important behavioral technique used in CBT, in which patients are guided to confront and tolerate conditions that provoke obsessions and compulsions and resist acting on them [4]. In the CBT program of this study, ERP accounted for nearly half of the treatment time (five out of 14 sessions). Based on the anatomical characteristics of VTN, we speculate that VTN and its related network might play an important role in the behavioral alteration during CBT involving ERP.

In this study, the comprehensive prediction model showed that the right entorhinal cortex thickness was related to OCD alleviation. Recently, one study investigating the cortical thickness in OCD patients, based on 780 brain scans from six centers, revealed that abnormal cortical thickness exists in broad areas. This finding partially supports the frontostriatal model of OCD, but also suggests that the limbic, temporal and parietal regions play a role in the disorder’s pathophysiology [36]. The entorhinal cortex (EC), located in the medial temporal lobe, is the central structure of memory formation and navigation. Traditionally, it has been regarded as a hub for information transmission and processing from the neocortex to the hippocampus [37]. One study, exploring the cortical morphological networks using cortical thickness, sulcal depth and surface area, found that the entorhinal cortex acts as a morphological ‘hub’ in the network [38]. Thus, it is worth further exploring the role of the entorhinal cortex in the CBT process.

### 4.4. Limitations

To our knowledge, this is the first study exploring the utility of pretreatment cortical features in predicting CBT outcomes in OCD patients using the SVR model. Although comprehensive cortical features display a strong predictive power for CBT, several limitations were identified. First, to ensure the predictive ability for CBT is OCD−specific, patients with comorbid conditions were excluded. However, at least 50% of OCD patients in the clinical setting have at least one underlying psychological illness [2]. As such, the predictive accuracy of our model may vary according to other patient comorbidities. Therefore, these results cannot be directly applied to common OCD patients. Second, the research was a non-controlled and open-label study. The placebo response of the participants was not resolved. Although the evaluator did not participate in the study, both the therapist and the patient knew the designated treatment. Accordingly, the clinical response may have been exaggerated due to expected bias. Thus, our findings need further validation using sham-controlled studies. Third, we used AAL3 and DK40 templates to extract cortical structural features, respectively, which might lead to biased results. Lastly, due to the relatively small sample size, the current results should be interpreted with caution. Future studies need to improve the research design, expand the sample size, combining more factors, and use more rigorous analysis methods to further verify the predictive effect of cortical structural features on the efficacy of CBT. For example, researchers can try the “multi-site canonical correlation analysis with reference + joint-independent component analysis” (MCCAR + jICA), which can precisely identify co-varying multimodal imaging patterns closely related to the reference, such as cognitive scores [39].

## 5. Conclusions

In summary, this study demonstrated that cortical structural feature parameters were associated with the efficacy of CBT against OCD. In particular, the results confirmed our hypothesis that integrating four cortical structural features is superior to using a single feature in CBT outcome prediction among OCD patients. These structural features exist in the complex loop connecting the cortex and subcortical region, indicating that CBT involves extensive brain neural networks. In the future, we will use a larger sample size and combine different brain biological markers, including functional MRI indicators, EEG, neurotransmitters, etc., to identify more powerful efficacy indicators for CBT. Nonetheless, our findings provide a new perspective on designing individualized treatment strategies for OCD patients.

## Figures and Tables

**Figure 1 brainsci-12-00921-f001:**
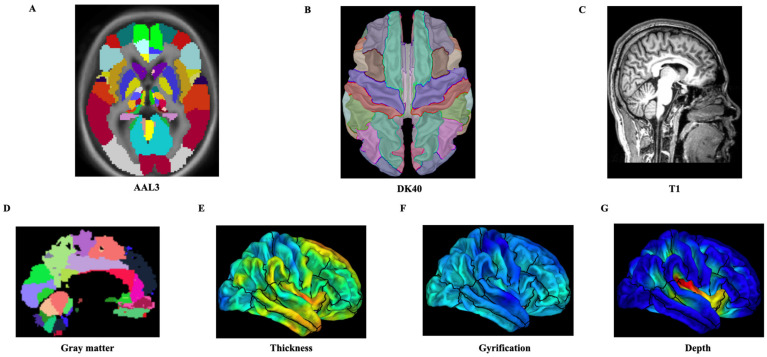
The extracted four cortical structure indices from the original MRI images based on different templates. (**A**) The template of AAL3; (**B**) the template of DK 40; (**C**) the raw MRI image of a subject; (**D**) the gray matter map of the subject with AAL3; (**E**) the cortical thickness of the subject with DK 40; (**F**) the gyrus of the subject with DK 40; (**G**) the sulcal depth of the subject with DK 40. (**A**,**B**) were shown by the Brainstorm; (**C**) was shown by MRIcron; and (**D**–**G**) were shown by the CAT12. Abbreviations: AAL3, automated anatomical labelling 3; DK 40, Desikan–Killiany atlas.

**Figure 2 brainsci-12-00921-f002:**
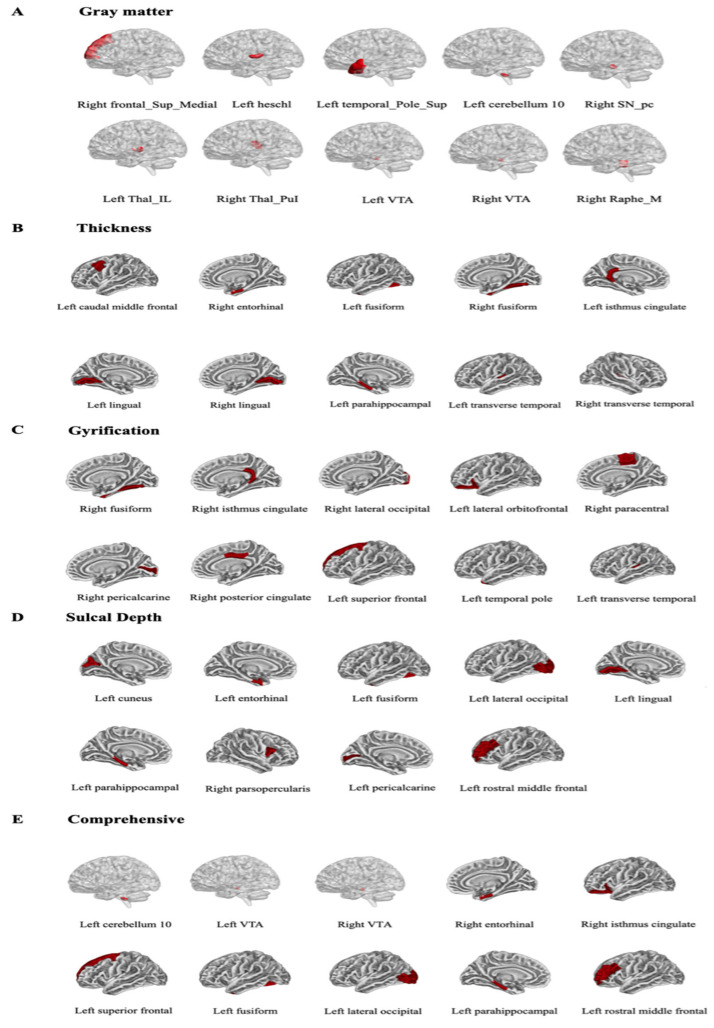
The distribution of the selected brain regions used in constructing the five prediction models. (**A**) the distribution of the gray matter volume; (**B**) the distribution of the cortical thickness; (**C**) the distribution of the gyrification; (**D**) the distribution of the sulcal depth; (**E**) the distribution of the comprehensive variables, including the left cerebellum 10, left VTA, and right VTA in the gray matter volume; right entorhinal is under the cortical thickness; right isthmus cingulate under the gyrification; Left superior frontal, fusiform, lateral occipital, parahippocampal, and rostral middle frontal regions are under the sulcal depth. Abbreviations: VTA, ventral tegmental area.

**Figure 3 brainsci-12-00921-f003:**
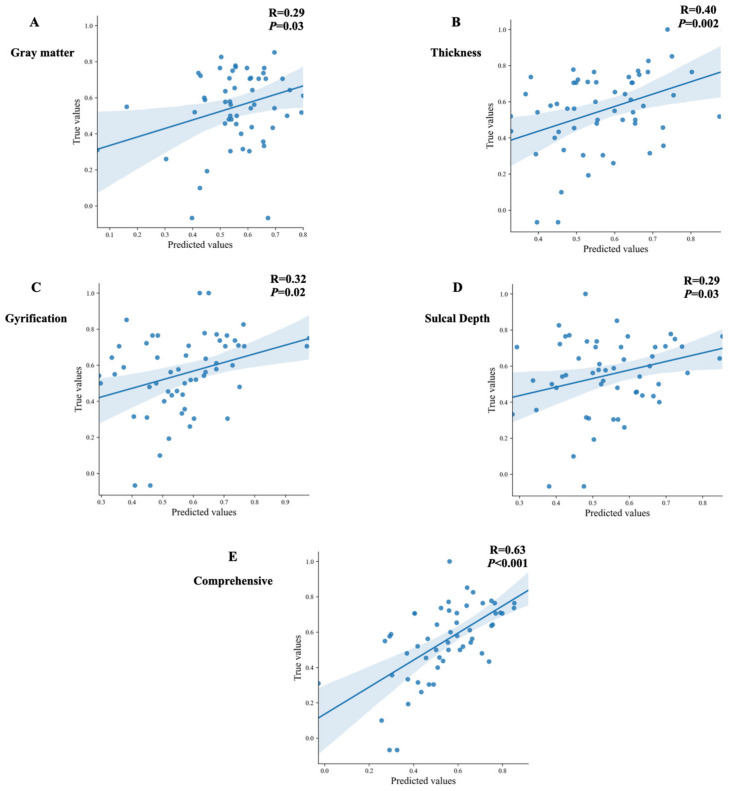
The correlations between true values and predicted values in the regressions of different features. (**A**) the correlations between true values and predicted values in the regressions with gray matter volume values; (**B**) the correlations between true values and predicted values in the regressions with thickness values; (**C**) the correlations between true values and predicted values in the regressions with gyrification values; (**D**) the correlations between true values and predicted values in the regressions with sulcal depth values; (**E**) the correlations between true values and predicted values in the regressions with comprehensive values.

**Figure 4 brainsci-12-00921-f004:**
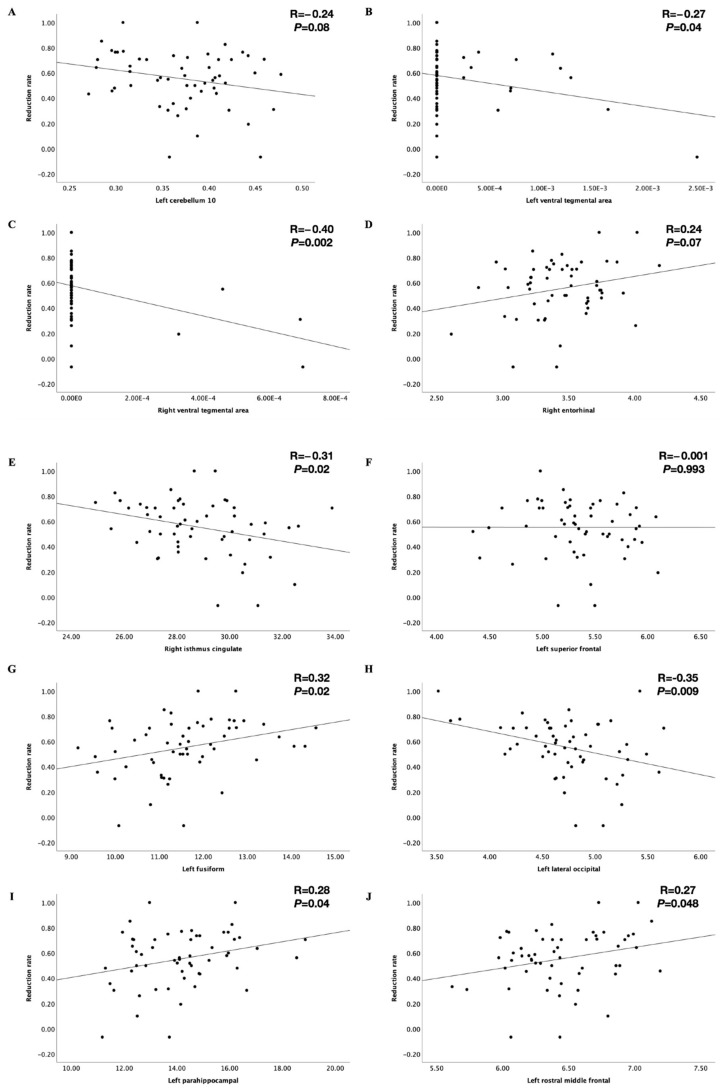
The correlation between the reduction rate of Y−BOCS score and comprehensive variables. (**A**–**C**) the correlation between the reduction rate of Y−BOCS score and gray matter volumes in left cerebellum 10, left VTA, and right VTA; (**D**) the correlation between the reduction rate of Y−BOCS score and thickness in right entorhinal; (**E**) the correlation between the reduction rate of Y−BOCS score and gyrus value in right isthmus cingulate; (**F**–**J**) the correlation between the reduction rate of Y−BOCS score and depth value in left superior frontal, fusiform, lateral occipital, parahippocampal, and rostral middle frontal regions.

**Table 1 brainsci-12-00921-t001:** The overview of the 14 CBT sessions.

Session	Main Technique	Week
The 1st–2nd session	Therapeutic alliance establishment, information collection and assessment, psychoeducation, and normalization.	The 1st week
The 3rd–6th session	Case conceptualization, identification of cognitive distortion, plan and implementation of behavioral experiments, and challenging and correction of the distorted cognitions, including on-site and homework exercises.	The 2nd–4th week
The 7th–12th session	Introduction of exposure, creation of an anxiety hierarchy and planning of exposure, and the conduct of ERP practice, including therapist-assisted and self-administered practice.	The 5th–10th week
The 13th–14th session	Treatment review to consolidate treatment effects and prevent relapse.	The 11th–12th week

Participants were not allowed to receive any psychoactive medications during the course of the study.

**Table 2 brainsci-12-00921-t002:** Demographic and clinical characteristics of OCD patients before and after CBT.

Characteristics	Baseline (*n* = 56)	12 Weeks (*n* = 56)	*p* Value (Paired−*t* Test)
Age	28.02 ± 6.7	--	
Gender (male/female)	37/19	--	
Education level (years)	15.5 ± 2.3	--	
Illness duration	10.35 ± 7.5	--	
Y−BOCS score			
Total	23.43 ± 5.84	10.68± 6.86	<0.001
Obsession	12.8 ± 4.6	5.3 ± 3.9	<0.001
Compulsion	11.3 ± 4.1	5.4 ± 3.2	<0.001
HAMD−17 score	6.29 ± 4.06	2.32 ± 2.83	<0.001
HAMA score	8.09 ± 5.89	2.61 ± 3.33	<0.001

Note: Data are expressed as mean ± SD, SD: standard deviation. OCD: obsessive–compulsive disorder. Y−BOCS: Yale–Brown Obsessive–Compulsive Scale. HAMD−17: the 17−item Hamilton Depression Rating Scale. HAMA: Hamilton Anxiety Rating Scale.

**Table 3 brainsci-12-00921-t003:** Regression performance for different structural feature representations.

Selected Features	Brain Regions	MAE	MSE	R^2^
GMV	Right superior frontal gyrus−medial Left Heschl’s gyrus Left temporal pole: superior temporal gyrus Left lobule X of the cerebellar hemisphere Left lateral geniculate Right anterior cingulate cortex−subgenual Left ventral tegmental area Right ventral tegmental area Right lobule III of vermis Right raphe nucleus–median	0.18	0.05	−0.02
Cortical thickness	Left caudal middle frontal Right entorhinal Left fusiform Right fusiform Left isthmus cingulate Left lingual Right lingual Left parahippocampal Left transverse temporal Right transverse temporal	0.17	0.04	0.12
Gyrification value	Right fusiform Right isthmus cingulate Right lateral occipital Left lateral orbitofrontal Right paracentral Right pericalcarine Right posterior cingulate Left superior frontal Left temporal pole Left transverse temporal	0.18	0.05	−0.03
Sulcal depth	Left cuneus Left entorhinal Left fusiform Left lateral occipital Left lingual Left parahippocampal Right parsopercularis Left pericalcarine Left rostral middle frontal	0.18	0.05	−0.02
Comprehensive variables		0.14	0.03	0.36
GMV	Left CER10 Left VTA Right VTA			
Cortical thickness	Right entorhinal			
Gyrification value	Right isthmus cingulate			
Sulcal depth	Left superior frontal Left fusiform Left lateral occipital Left parahippocampal Left rostral middle frontal			

Note: MAE: Mean absolute error; MSE: Mean squared error; R^2^: Coefficient of determination; GMV: Gray matter volume; CER: Cerebellum; VTA: Ventral tegmental area.

## Data Availability

Not applicable.

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
