# Peer review of "Comprehensive Cortical Structural Features Predict the Efficacy of Cognitive Behavioral Therapy in Obsessive-Compulsive Disorder"

_brainsci, 2022, doi:10.3390/brainsci12070921_

Round 1

Reviewer 1 Report

Undoubtedly, cognitive behavioral therapy is effective for patients with obsessive-compulsive disorder, but still 40% patients with OCD do not have good respondence to CBT. Authors correctly suggests that future research needs to comprehensively consider various brain indicators, which will help to find more powerful efficacy predictors for CBT.

My comments to the article are as follows:

- I propose to extend the background in the field of methods of acquisition and archiving of data from the human brain by quoting, for example: Methods of acquisition, archiving and biomedical data analysis of brain functioning, Biomedical Engineering and Neuroscience, Proceedings of the 3rd International Scientific Conference on Brain-Computer Interfaces, BCI 2018, March13-14, Opole, Poland, Advances in Intelligent Systems and Computing book series (AISC, volume 720).

- I would suggest dividing Figure 1 into smaller sections, interspersed with descriptive text.

- In my opinion, Conclusions should be extended to include plans for the future in the field of research.

Author Response

Thank you very much for your careful review. Your comments have greatly improved our article. Please check our point- to- point reply in the attachment.

Reviewer 2 Report

Here are some suggestions to help improve the paper.

1)There is a sentence in the abstract that says “The results show that 28 cortical structural features are associated with the efficacy of CBT in OCD patients.” However, the authors need to present data and correlations that give evidence for the association between cortical structural features and efficacy of CBT. The correlations and figure with true value versus predicted value are very helpful; however, a correlation also needs to be presented with X = cortical structure score (combined or single) versus Y = pre-post CBT clinical score in order to give evidence for sentence presented above in the abstract.  Present a scatter plot using X axis = cortical structure score (combined or single) versus Y axis = pre-post CBT clinical score complete with regression line fit.

2) there are some minor typos that need to be corrected for example

This sentence “First, a total of 376 the structural features were extracted based on the atlas.”

should be changed to

“First, a total of 376 structural features were extracted based on the atlas.”

3) Add figures that show an example of the raw MRI image and the structural features extracted such as GMV, cortical thickness, sulcal depth superimposed on an image so the reader can see the parameter being measured.

Author Response

Thank you very much for your careful review. Your comments have greatly improved our article. Please see our point-to-point reply in the attachment.

Reviewer 3 Report

The author uses structural MRI in this article to predict the effectiveness of CBT treatments for patients suffering from obsessive compulsive disorders. The objective of the article is to evaluate the capability of features (GMV, cortical thickness, sulcal depth and gyri value) from structural MRI in predicting the effectiveness of CBT in patients with obsessive-compulsive disorders. The article addresses an interesting topic of research since structural imaging combined with machine learning is becoming more and more popular as a technique for predicting treatment outcomes. It is also interesting that in this paper, unlike other papers that try to find diagnosis using machine learning, they are attempting to find prediction value in structural imaging for treatments, and this will have a great impact for clinical applications. This is quite an interesting paper, the method used and the results are of some clinical significance, but there are some limitations, which I will present here. 

First of all, there is a limitation of a small sample size, which can constitute a problem when we try to apply machine learning tools that, as we know, require substantial larger  data sets. Because the authors compared different methods and features, the model parameters grew, and there is a need for additional data. I believe the author should explain this limitation to the reader and place a word of caution before taking any generalizations from the analysis.   I still believe that the authors' comparison of different methods of extracting features is a valuable aspect of the analysis of data. They focus on comparing the different features and their ability to predict better.  Nevertheless, the features in the method are quite different, and it's not clear why one atlas is used for gray matter and another for cortical thickness, creating an unnecessary variable.  Furthermore, I think that the author should perform post hoc analyses with their analyses and conduct a "fairness" analysis, first to determine whether there are any biases in age, if younger participants do better or worse in term of prediction accuracy compared to older adults and similarly with potential gender biases. The analysis can also be applied to psychiatric measures.   Lastly, the authors should explain why the predictive regions differ so much from model to model, and they should discuss some means of combining the classifiers from the different models/features. 

Author Response

(The authors gave the same response as above.)

Round 2

Reviewer 2 Report

The authors have revised the manuscript and have addressed all of my previous concerns.